# Development of Biomaterials to Modulate the Function of Macrophages in Wound Healing

**DOI:** 10.3390/bioengineering11101017

**Published:** 2024-10-12

**Authors:** Jiacheng Li, Jiatong Xie, Yaming Wang, Xixian Li, Liqun Yang, Muxin Zhao, Chaoxian Chen

**Affiliations:** 1Department of Plastic Surgery, The Second Affiliated Hospital, Dalian Medical University, Dalian 116041, China; lijiacheng200710@163.com (J.L.); lianjinyuansu@foxmail.com (X.L.); 2The Second Clinical College, Dalian Medical University, Dalian 116044, China; x2816257241@163.com; 3The First Affiliated Hospital, Dalian Medical University, Dalian 116014, China; dlwym@mail.dlut.edu.cn; 4Research Center for Biomedical Materials, Engineering Research Center of Ministry, Education for Minimally Invasive Gastrointestinal Endoscopic Techniques, Shengjing Hospital of China Medical University, Shenyang 110022, China; yanglq@lnszjk.com.cn; 5School of Materials Science and Engineering, Key Laboratory of Polymer Chemistry and Physics of Ministry of Education, Peking University, Beijing 100871, China

**Keywords:** macrophage, wound healing, biomaterial

## Abstract

Wound healing is a complex and precisely regulated process that encompasses multiple stages, including inflammation, anti-inflammation, and tissue repair. It involves various cells and signaling molecules, with macrophages demonstrating a significant degree of plasticity and playing a crucial regulatory role at different stages. In recent years, the use of biomaterials, which include both natural and synthetic polymers or macromolecules, has proliferated for the purpose of enhancing wound healing. This review summarizes how these diverse biomaterials promote wound healing by modulating macrophage behavior and examines the broader implications of these modulations. Additionally, we discuss the limitations associated with the clinical application of immunomodulatory biomaterials and propose potential solutions. Finally, we look towards future developments in the design of immunomodulatory biomaterials intended to enhance wound healing.

## 1. Introduction

Skin wounds are a pathological condition resulting from various causes, including disease, injury, or exposure to physical and chemical irritants. They can be categorized based on the nature of the injury and the expected healing timeline, distinguishing between acute wounds, which typically heal within a short, defined period, and chronic wounds, which may persist and heal over an extended period due to various underlying factors [1]. Acute wounds typically arise from physical or chemical trauma and surgical procedures. In contrast, chronic wounds are often precipitated by underlying infections or diseases such as diabetes and vascular disorders [2,3].

Studies have demonstrated that macrophages exert a crucial influence on the wound-healing process [4]. The plasticity of macrophages is essential for wound repair [5]. Macrophages can polarize into M1 (pro-inflammatory) and M2 (anti-inflammatory) phenotypes, with M1 types involved in pathogen phagocytosis and the clearance of damaged cells, including neutrophils [6], and with M2 types contributing to reparative and regenerative processes [7]. The transition from M1 to M2 polarization in macrophages indicates a change from a pro-inflammatory state, characterized by the production of inflammatory cytokines and reactive oxygen species, to an anti-inflammatory and tissue repair state, in which M2 macrophages promote healing and the resolution of inflammation [8]. Biomaterial-based strategies for wound closure typically aim to facilitate re-epithelialization, the process by which the wound bed is resurfaced with new epithelium, and to support macrophage-mediated tissue repair and function restoration, which are distinct yet complementary phases of the wound-healing continuum [9].

Recent advances in wound-healing research and clinical applications can be attributed to the development of immunomodulatory biomaterials, which have demonstrated the capacity to modulate immune responses and improve the recovery function of tissue, thereby significantly improving outcomes in wound care [10]. Biological materials are versatile components in wound healing, functioning as scaffolds or dressings. They are capable of incorporating a multitude of bioactive substances or cells, thereby constructing an integrated wound repair system that synergistically delivers cells and growth factors to the site of injury [11]. To address complex wound infections, such as those encountered in diabetes and chronic inflammation, along with issues of vascularization deficiency, there is an urgent need to develop biomaterials with multifunctional properties, including antibacterial activity, immune regulation, and angiogenesis promotion. These materials are essential for the comprehensive enhancement of the wound-healing process [12,13].

Biomaterials research emphasizes immunomodulation, in which these materials are engineered to ameliorate chronic inflammation, rectify immune dysregulation, and expedite tissue repair [14]. For instance, specific biomaterials are designed to facilitate the transition from the inflammatory to the proliferative phase of wound healing by modulating macrophage polarization [15]. In addition to immune regulation, these biomaterials can also encapsulate and deliver growth factors, including vascular endothelial growth factor (VEGF) and fibroblast growth factor (FGF), to stimulate angiogenesis and cell proliferation within the wound [16].

## 2. Classification of Immunomodulatory Biomaterials

Immunomodulatory biomaterials are defined as materials that can regulate immune responses to control inflammation and promote tissue repair, thus creating a favorable wound regeneration microenvironment [17]. As a result, immunomodulatory biomaterials are expected to influence immune cell function, which may contribute to enhanced tissue healing [18]. Among them, the main immunomodulatory biomaterials are categorized into two principal classes: natural polymers and synthetic polymers [10], as shown in Figure 1.

Natural polymers are readily accessible, simple to prepare, exhibit minimal levels of immunogenicity and low levels of toxicity, and possess favorable levels of biocompatibility [19]. Natural materials utilized in wound healing encompass a variety of proteins, including collagen [20], silk fibroin [21], and gelatin [1]. Additionally, polysaccharides such as hyaluronic acid [22], sodium alginate [4], and chitosan [23] are also commonly applied for their beneficial properties in the healing process. For example, polysaccharide-based biopolymers, which are extensively utilized in tissue engineering, drug delivery, and bioelectromechanical systems, can be tailored as immunomodulatory agents to enhance wound healing and tissue regeneration [24]. Synthetic polymers used in the realm of wound healing include polyesters such as polylactic acid (PLA) [25], polyglycolic acid (PGA) [26], and poly-caprolactone (PCL) [27]. Other classes of polymers encompass polyanhydrides, exemplified by polyamino acid [28], polyurethane [29], and various other polymers. Polylactic acid-co-glycolic acid (PLGA) is one of the most studied synthetic polymers. PLGA-based wound dressings have garnered significant attention for their ability to create a moist environment that facilitates autolytic debridement and promotes the progression of the wound-healing phase [30]. Synthetic polymers offer a high level of controllability in their design and synthesis, which can effectively compensate for the limitations inherent in natural materials. However, the variability in their physical and chemical properties can be challenging to control, and the range of available options is often limited. This can affect the precision of their integration with tissue repair processes and potentially hinder optimal wound-healing outcomes.

Accordingly, to leverage the advantageous characteristics of materials, such as their biocompatibility, degradation rate, and mechanical strength, while addressing their inherent limitations, researchers are exploring the combination of different materials to create composites that can potentially enhance wound-healing effectiveness [31]. Composites possess intrinsic properties that facilitate the wound-healing process. The cellular constituents and bioactive molecules engaged in this reparative procedure are influenced by a variety of extrinsic factors, including physical and mechanical signals [20], chemical stimuli, and the intrinsic properties of the material [32]. These factors collectively modulate cellular behavior, thereby orchestrating the complex sequence of events that lead to tissue regeneration and repair [33]. Nanocomposites are one of the most popular composites. Polymer-based nanocomposites, such as PLGA nanoparticles, as efficient drug delivery systems, enable the control of the release of growth factors and antibiotics, which is essential for initiating and sustaining the wound-healing cascade [34]. Nonetheless, while various synthetic materials used for wound healing may exhibit pronounced shortcomings, such as poor biocompatibility or unsuitable degradation rates, the creation of blended and composite materials can significantly enhance their performance. By combining these synthetic polymers with other materials, we can leverage the high level of controllability of synthetics to tailor their properties, effectively addressing their inherent limitations [35]. This blending approach expands the range of available options and allows for a more precise tuning of physical and chemical properties to better integrate with tissue repair processes, thus potentially improving wound-healing outcomes [36].

Furthermore, the enhancement of material performance and the optimization of the composition ratio within composite materials can lead to improved biocompatibility and more controlled degradation rates. These advances are crucial for the development of effective carriers in an exogenous drug delivery system, as they allow for the more precise tuning of the material properties to specific requirements for drug delivery and tissue repair processes [37]. By fine-tuning the composition of composite materials, we can create carriers that provide a more favorable environment for drug release and integration with the biological system, potentially enhancing the overall efficacy of wound healing and tissue regeneration. Also, advanced material processing techniques can be used to refine the material’s internal structure, which can improve the delivery of therapeutic agents and thus enhance the healing process [38]. For instance, a novel medical adhesive derived from natural polysaccharides extracted from snail mucus secretions exhibits superior adhesion properties compared to traditional fibrin glue and facilitates wound healing [39]. Guan et al. have developed a biomimetic lotus thread bacterial cellulose hydrogel fiber, which replicates the helical structure of lotus root fibers, endowing it with exceptional toughness and strength. This innovative material is poised for application in high-end surgical sutures. These sutures are designed to support the body’s natural healing processes by providing a framework for tissue regeneration and minimizing the risk of complications during wound healing [40].

## 3. Wound Healing

Skin wound healing is a protracted and dynamically regulated process, typically characterized by four sequentially occurring yet overlapping stages: hemostasis, inflammation, proliferation, and remodeling [41], as shown in Figure 2. Hemostasis is an immediate response to injury, characterized by the rapid constriction of blood vessels and the activation of platelets, culminating in the formation of fibrin clots. These clots serve multiple functions: they prevent the ingress of bacteria, release complement proteins and growth factors, and provide a temporary extracellular matrix that facilitates the infiltration of cells essential for the subsequent phases of wound healing [6,41].

The inflammatory phase is primarily characterized by the activation of the immune system, which is tasked with the clearance of pathogens at the site of injury [42]. Post-injury, neutrophils and macrophages are actively drawn to the wound site by the reactive oxygen species (ROS) emitted from the damaged tissue, representing an active mechanism of the body’s response to injury [43]. Following this initial attraction, macrophages play a subsequent role by secreting proinflammatory cytokines, such as interleukin (IL)-6, tumor necrosis factor (TNF)-α, and IL-1β, which further amplify the inflammatory response. Additionally, macrophages utilize ROS for pathogen neutralization, contributing to the clearance of potential infections at the wound site [44]. They release monocyte chemotactic protein (MCP)-1 and matrix metalloproteinases (MMPs) to recruit additional immune cells and facilitate extracellular matrix (ECM) degradation, respectively. The presence of damage-associated molecular patterns (DAMPs), recognized by macrophage pattern recognition receptors (PRRs), sustains inflammation through Toll-like receptors (TLRs) and inflammasome activation [45]. The clearance of apoptotic neutrophils by macrophages via engulfing indicates the onset of the resolution phase in wound healing [6,46,47]. In the proliferative phase, granulation tissue forms de novo, independent of angiogenesis, which is a process specific to endothelial cells. Angiogenesis, the growth of new blood vessels, occurs subsequent to the formation of granulation tissue and is driven by endothelial cell activity. Fibroblast proliferation and differentiation into myofibroblasts are key processes that facilitate wound contraction and significantly contribute to tissue repair. This sequence of events highlights the distinct yet complementary roles of granulation tissue and angiogenesis in the wound-healing process [48]. Proliferating fibroblasts secrete components into the extracellular matrix, thereby altering the wound microenvironment from a predominantly inflammatory state to one conducive to tissue growth and repair [42]. In the proliferative phase, as inflammation subsides and granulation tissue forms, macrophages polarize from M1 to M2 phenotypes, a pivotal change that promotes the transition from inflammation to tissue repair, critically affecting the quality of wound healing [6]. Macrophages release growth factors such as VEGF and platelet-derived growth factor (PDGF), initiating angiogenesis by activating endothelial cells. They also participate in extracellular matrix (ECM) deposition, with a subset potentially leading to fibrosis, known as fibrocytes or classified as the M2a macrophage subtype [49].

As the wound enters the remodeling phase, macrophages revert to their original phenotype (M0), releasing proteases that degrade excess cell debris and the extracellular matrix (ECM). As the wound-healing process nears completion, the initial granulation tissue, which is highly cellular and vascular, is gradually replaced by a more mature collagen-rich tissue with fewer cells. This transition signifies the final phase of wound healing, resulting in a scar that has a lower cell count and is characteristic of the healed state [47]. Macrophages are essential for normal wound healing and tissue regeneration. Increasing the number of monocytes or macrophages in the wound can significantly accelerate wound healing in normal and diabetic mice [4].

## 4. Design of Immunomodulatory Biomaterials to Promote Wound Healing—Mechanism of Action

In the domain of tissue repair, harnessing immune modulation to foster tissue regeneration presents an innovative therapeutic approach for addressing chronic wounds and extensive tissue injuries. Our growing understanding of the intricate interplay among immune cells, stem/progenitor cells, and other cellular constituents pivotal to the M1-to-M2 macrophage polarization transition is propelling significant advancements in this field [46]. Material-based strategies encompass modulating surface chemistry and topography, as well as the incorporation of nanoparticles (NPs) to direct macrophage behavior. Nanometer-scale materials, characterized by distinct chemical compositions and surface properties, can induce a regenerative or anti-inflammatory macrophage phenotype without the need for growth factors or cytokines [47,48], as shown in Figure 3.

Additionally, the topological features of biomaterials can influence macrophages’ cytoskeletal structure, thereby regulating their polarization [49]. Aligned electrospun nanofibers can influence macrophage polarization via the JAK-STAT and NF-κB pathways and attenuate local inflammatory response in skin wounds. The covalent attachment of chemical moieties on biomaterials can attenuate the pro-inflammatory M1 macrophage response, promoting a shift towards the M2 phenotype [50]. In the rat experimental model, the scaffolds functionalized with covalently bound sulfated hyaluronan (sHA3 + EDC/NHS) demonstrated a reduced activation of pro-inflammatory M1 macrophages, which was evidenced by the diminished recruitment and activation of pro-inflammatory cells [51]. The mechanical properties of scaffolds, such as their stiffness, can also modify the polarization of macrophages, influencing the balance between the pro-inflammatory M1 phenotype and the anti-inflammatory M2 phenotype. This is important because macrophage responses are not solely determined by the mechanical properties of the scaffolds but are part of a broader microenvironmental sensing and response mechanism that is crucial for promoting the wound-healing process towards tissue regeneration [52]. Guo et al. designed poly(ester urethane) scaffolds with mechanical properties that can facilitate cell infiltration, collagen deposition, angiogenesis, Wnt signaling pathway activation, and macrophage polarization, which have been found to promote tissue regeneration and repair [53]. Findings indicate that interactions between macrophage mannose receptors (MR) and specific antibodies, leading to the formation of nanoscale clusters, can facilitate the transition of macrophages from the M1 to M2 phenotype. After this, the existing body of research was systematically evaluated to elucidate the specific immunomodulatory functions and underlying mechanisms through which these biomaterials exert their therapeutic effects on wound repair.

### 4.1. Influence of Physical Properties of Biomaterials on Macrophage Behavior

The innovative design of biomaterials, particularly their pore structure, offers a critical physical scaffold that facilitates macrophage migration and infiltration within the wound bed [50,51]. Surface roughness significantly influences macrophage adhesion, proliferation, and maturation processes, which are pivotal for effective tissue integration and healing [52]. Moreover, the mechanical stiffness of materials is a determinant factor for macrophage polarization [53], dictating their functional roles and migration patterns. The Young’s modulus of scaffolds is of paramount importance, as it significantly influences cellular behavior, including adhesion, proliferation, and differentiation [54]. Polyethylene glycol (PEG), known for its low Young’s modulus and excellent flexibility, is frequently utilized in hydrogel formulations and drug delivery systems to enhance cell compatibility and facilitate tissue repair [55]. Poly(lactic-co-glycolic acid) (PLGA), a copolymer with a tunable mechanical strength and Young’s modulus, is extensively applied in drug delivery systems and tissue engineering scaffolds. Clinically, it is employed in the fabrication of degradable fracture fixation screws and bone tissue engineering scaffolds [56].

Furthermore, biomaterials with a rough, hydrophilic surface can foster an anti-inflammatory microenvironment, thereby enhancing the healing response to the implant [57,58]. Polylysine-grafted poly(propylene fumarate) polyurethane films (PPFU) can restrict M1 polarization, whereas they promote the M2 polarization of macrophages in vitro [59]. Wang et al. demonstrated that stent architecture can modulate macrophage polarization: thicker-fiber electrospun PCL vascular grafts could enhance the process of vascular regeneration and remodeling by mediating macrophage polarization into the M2 phenotype [60]. Almeida et al. reported that chitosan scaffolds, characterized by larger pores and wider angles compared to PLA stents, resulted in an elevated production of proinflammatory cytokines [61]. The study also highlighted that by fine-tuning the three-dimensional platform’s properties, encompassing its chemistry, morphology, and framework, macrophage responses can be effectively regulated. McWhorter et al. have shown that the polarization of macrophage phenotypes is critically influenced by the cellular microenvironment, indicating that modulating the physical properties of this environment can precisely regulate immune cell behavior [62]. The cytoskeleton plays a role in the complex interactions between macrophages and the microenvironment’s physical and chemical factors. These interactions are facilitated by adhesion proteins, particularly integrins, on the cell surface. Integrins are involved in sensing the physical properties of the extracellular matrix (ECM) and mediating the indirect interactions between the ECM and the cell. This process is often mediated by molecular species, such as serum glycoproteins, which can bind to both the material surface and cell surface receptors, creating a bridge that allows for signal transduction into the cell [63].

Immune-active nanofiber scaffolds have been developed to modulate macrophage activity, facilitating the transition from the pro-inflammatory M1 phenotype to the anti-inflammatory M2 phenotype, thereby accelerating wound healing [64]. Similarly, fish scale scaffolds, when loaded with mesenchymal stem cells (MSCs), effectively attenuate the inflammatory response by inducing a shift in macrophage polarization towards the M2 phenotype, leading to reduced inflammation in the surrounding tissue [65]. Luo et al. demonstrated in a rat skin injury model that the decellularized extracellular matrix (dECM), enhanced by three freeze-thaw cycles, significantly induced M2 macrophage polarization [32], which suggested that the immunomodulatory property of the dECM can be efficiently manipulated by tailoring its inherent micromechanical properties during the decellularization process.

### 4.2. Influence of Material Chemical Properties on Macrophage Behavior

The chemical properties of biomaterials, particularly those that come into direct contact with immune cells such as macrophages, play a pivotal role in modulating immune response and the overall wound-healing process. Surface modifications, such as RGD peptide immobilization on polymers, augment macrophage adhesion and drive polarization toward phenotypes conducive to tissue regeneration [66]. The RGD peptide is a small molecular mimetic of specific sequences within the extracellular matrix (ECM). It selectively binds to integrin receptors on the cell surface and enhances the adhesion capability of macrophages [67]. Additionally, polyethylene glycol (PEG), with its hydrophilic properties, promotes optimal conditions for cell culture while mitigating inflammatory reactions. The coating of biomaterials with PEG has demonstrated the ability to reduce protein adsorption and subsequent macrophage activation, leading to a more controlled inflammatory response [68,69]. Glycosaminoglycans (GAGs) possess a high degree of hydrophilicity, enabling them to attract water and provide a conducive microenvironment for cells, which aids in cellular adhesion and migration [70]. Glycosaminoglycan grafting on biomaterials, such as hyaluronic acid, modulates macrophage polarization to the M2 phenotype, promoting tissue repair and angiogenesis [71]. Biomaterial-based drug delivery systems enable the precise regulation of macrophage activity by controlling the release of immunomodulatory agents, such as the sustained delivery of anti-inflammatory drugs, which can optimize the wound healing environment [72]. Recent progress underscores the pivotal role of biomaterial chemistry in crafting immunomodulatory scaffolds, necessitating tailored designs that account for macrophage responses.

### 4.3. Biological Properties of Biological Materials’ Impact on Macrophage Behaviors

During the initial phase of wound healing, the predominant macrophage phenotype is M1 [8], characterized by its pro-inflammatory role. These cells are instrumental in the clearance of cellular debris and residual neutrophils, which is essential for preventing infection and facilitating the progression of the healing process [73]. M1 macrophages actively engage in phagocytosis, a critical process for the removal of dead cells and pathogens from the wound site. Concurrently, they secrete a spectrum of pro-inflammatory cytokines, including IL-1β, IL-6, and TNF-α [74]. These cytokines are pivotal in initiating the inflammatory response, which is necessary for recruiting additional immune cells to the site of injury, activating the coagulation cascade, and initiating subsequent phases of tissue repair [46].

#### 4.3.1. Macrophage Polarization

Biomaterials can influence macrophage polarization, steering them from the pro-inflammatory M1 to the anti-inflammatory M2 phenotype, thus affecting inflammation and tissue repair [75]. They achieve this by modulating key signaling pathways: inhibiting TLR4/NF-κB to suppress M1 polarization [76] and promoting M2 polarization [22,77]. The NF-κB pathway, involved in mechanosensing, also plays a role in polarization by enhancing cell adhesion [32]. Moreover, the JAK-STAT pathway contributes to M1 inhibition, while the IL-4/Stat6 pathway facilitates the M2 transition [23,78].

##### Natural Polymers

Hydrogel materials have made significant strides in various biomedical applications due to their ability to mimic the dynamic characteristics of the ECM through their highly hydrated microenvironments, enabling advances in tissue regeneration, drug delivery, and disease modeling [79]. Hydrogel materials employ strategies to modulate macrophage polarization. A hydrogel can convert M1 macrophages into M2 macrophages without the need for any additional ingredients or external interventions [1]. Wang et al. discovered a snail mucus-derived polysaccharide binder (d-SMG) that promotes macrophage polarization to the M2 phenotype, leading to reduced chronic wound inflammation and improved healing via the STAT3 pathway [39]. Fu et al. reported that hybrid collagen-based all-natural hydrogels incorporating protocatechualdehyde possess the unique ability to induce the phenotypic transition of M1 macrophages to M2 macrophages [1]. This modulation of macrophage heterogeneity, along with the promotion of angiogenesis, significantly contributes to the healing of diabetic wounds. The GM-P@HA-P hydrogel has demonstrated the ability to effectively modulate macrophage polarization towards the M2 phenotype by activating the mannose receptor (MR) and the ERK/STAT6 signaling pathway, which in turn promotes the resolution of inflammation and enhances chronic wound healing [80]. This is evidenced by the significant reduction in CD86-positive macrophages and the concurrent increase in CD206-positive macrophages at the wound sites. In a recent study by Zhang et al., the incorporation of prostaglandin E2 (PGE2) into chitosan (CS) hydrogel was shown to effectively promote the M2 polarization of macrophages through sustained PGE2 release [81]. This CS + PGE2 hydrogel-mediated M2 transition is associated with reduced inflammation and enhanced tissue repair [81].

In inflammation, miR-29ab1 (microRNA-29a and microRNA-29b1) plays a crucial role, with miR-29ab1 knockout mice exhibiting enhanced wound healing and reduced inflammation. The newly developed Chitosan@Puerarin (C@P) hydrogel inhibits miR-29ab1, curbing M1 macrophage polarization and inflammatory response in wounds, thus promoting wound repair [82]. In in vitro experiments, a CS + PGE2 hydrogel significantly reduced the accumulation of M1-type macrophages at the wound site and significantly reduced levels of local pro-inflammatory cytokine IL-1β at the wound site [81]. The Epigallocatechin gallate (EGCG)-Cu@CBgel system redirects macrophage metabolism, suppressing M1 polarization and pro-inflammatory cytokine expression, thereby precisely modulating the healing process in vitro and in vivo [83]. Fu et al. introduced a biocompatible flat patch with specifically designed holes that form a human dermal fibroblast sheet (SIS) incorporating spheroids to mediate the activity of inflammatory cytokines for M2 polarization and increase angiogenic efficacy [51].

##### Synthetic Polymers

Reactive oxygen species (ROS)-scavenging hydrogels modulate wound inflammatory milieu by neutralizing excess ROS, balancing inflammatory cytokines, and enhancing M2 macrophage polarization, thereby accelerating the healing of wounds [38,53,83]. Patil et al. discovered that treatment with polythioketal (PTK) urethane (UR) in porcine skin wounds resulted in a significant increase in CD206+ macrophages, which can reduce M1 macrophages and promote the polarization of M2 macrophages [84]. PtCuTe nanosheets mitigate inflammation, curtail pro-inflammatory cytokine release, and enhance M2 macrophage polarization [85]. The polyurethane–hyaluronic acid (PUHA) composite hydrogel scaffold facilitates the macrophage transition from the M1 to M2 phenotype, lowering inflammation and reshaping the immune microenvironment through cytofactor–cytokine receptor pathways to enhance wound healing [86].

##### Composites

One of the most promising composite materials is the nano zinc oxide–biomimetic nanocomposite (nZnO-BNC), which has shown significant potential in promoting wound healing. The hydrophilic surface of nZnO-BNC materials may influence the polarization of macrophages. Hydrophilic and anionic surfaces could potentially enhance the apoptosis of inflammatory macrophages, mitigate the inflammatory response, and decrease the formation of foreign body giant cells [87]. The hybrid biomaterial Gel@fMLP/SiO2-FasL, integrating formyl-met-leu-phe (fMLP) and FasL-conjugated silica nanoparticles within the pH-responsive hydrogel, leverages the FasL-Fas pathway to induce neutrophil apoptosis and macrophage phagocytosis, thereby facilitating the shift from M1 to M2 macrophage polarization [50]. Chitosan, a natural polycationic polymer, has been combined with other polymers and nanomaterials to create advanced wound dressings. These composites not only exhibit antimicrobial properties but also stimulate the activation of macrophages, supporting the healing process [88]. The high reactivity of chitosan, due to its abundance of hydroxyl and carboxyl groups, makes it an ideal candidate for the design of encapsulation and drug delivery systems [89].

#### 4.3.2. Utilizing Bioactive Factors for Immune Cell Modulation

Bioactive factors, such as protease inhibitors, cytokines, microRNAs (miRNAs), and small interfering RNAs (siRNAs), are employed to selectively target and modulate specific immune responses [90]. IL-10, serving as an anti-inflammatory and anti-fibrotic cytokine, exerts concentration-dependent effects on macrophage polarization and function [91].

##### Natural Polymers

Hydrogel-based systems for the delivery of cells, proteins, growth factors, and immune-regulatory substances have advanced our approach to modulating macrophage polarization for tissue repair. FDM (fibroblast-derived matrix)-gel effectively curtailed proinflammatory cytokine production and stimulated M2 macrophage activation [92]. Shen et al. introduced a novel hydrogel composite of sulphonated chitosan (SCS) and type I collagen (Col I/SCS) [23]. The Col I/SCS hydrogel has been shown to decrease levels of pro-inflammatory cytokines like IL-6 while enhancing the production of anti-inflammatory cytokines such as IL-4 and TGF-β1. It effectively reduces M1-like macrophage polarization and markedly accelerates the healing of diabetic wounds [80]. Mechanistic insights revealed that FDM-gel and Col I/SCS hydrogel interaction with macrophage integrins α5β1 and α1β1 induced VEGF and bFGF upregulation, Akt phosphorylation, and enhanced MMP-9 activity, collectively highlighting the essential role of macrophages in FDM-gel-mediated wound repair [23,92]. Zhang et al. developed a hydrogel, combining PRP with nano clay, which facilitated the sustained release of growth factors and deferoxamine to enhance M2 macrophage polarization, curb inflammation, and expedite wound healing in type 1 diabetic and normal rats, presenting a novel strategy for diabetic wound treatment [93].

##### Synthetic Polymers

Electrospun fibers have been shown to facilitate wound healing by releasing IL-10 in a controlled manner, as demonstrated by studies that detail their ability to modulate the immune response at the wound site [94]. In the early inflammatory phase, these fibers release a low dose of IL-10, which encourages an M2c macrophage response. M2c macrophages, a subtype of M2 macrophages characterized by their expression of specific surface markers and secretion of cytokines such as IL-10, play a role in curbing inflammation [91]. Poly(ethylene glycol) (PEG) hydrogel was utilized for the sustained release of exosome, effectively guiding macrophage polarization from the pro-inflammatory M1 to anti-inflammatory M2 phenotype during the transition from the inflammatory to proliferative phase [95]. A biomimetic hyaluronic acid (HA) hydrogel integrated with a pH-responsive H2S (hydrogen sulfide) donor, JK1, was developed to create an innovative HA-JK1 hybrid system for localized therapy. The HA-JK1 treatment group demonstrated an elevated M2 polarization level, with a significant increase in M2 phenotypic marker expression in macrophages and a reduction in inflammatory cytokine secretion [96]. The treatment also modulated macrophage antigen presentation, enhancing T cell activation and differentiation, thereby boosting the local immune response.

##### Composites

Composite materials with bioactive integrations are a promising strategy for modulating immune factors to enhance efficient wound repair. Chitosan-based composites, when combined with bioactive factors such as growth factors or cytokines, can modulate immune response and promote tissue regeneration. For instance, chitosan nanoparticles loaded with TGF-β have been shown to enhance wound healing by promoting the proliferation and migration of fibroblasts and keratinocytes [88]. Polyurethane is a versatile material that can be combined with ZnO nanoparticles to create a composite material with antimicrobial properties [97]. The addition of bioactive factors such as silver nanoparticles or antimicrobial peptides can further enhance the ability of the material to modulate the immune response and promote wound healing [87]. The synergistic effect of ZnO with bioactive factors enhances the efficacy of wound management by directly targeting infection and inflammation, thereby potentially influencing clinical practices through improved patient outcomes and reduced treatment durations [98]. When gelatin is combined with nanofibers, such as those made from PCL, the resulting composite material can be used to deliver bioactive factors like keratinocyte growth factor (KGF), which can stimulate epithelialization and improve the healing of deep wounds [99]. PLGA scaffolds can be loaded with bioactive factors such as bone morphogenetic proteins (BMPs) or insulin-like growth factor (IGF) to modulate the immune response and promote tissue repair. The controlled degradation of PLGA allows for the sustained release of these factors, enhancing the healing process [100].

## 5. Challenges and Prospects

The development of immunomodulatory biomaterials for wound healing has made significant strides, offering promising solutions to enhance the body’s natural healing process. However, designing new and more effective biological materials to promote tissue repair and regeneration needs a deeper understanding of the complex mechanism of control of these processes; challenges remain in optimizing these materials to achieve consistent and effective outcomes across different wound types and patient populations.

The physical properties of biomaterials, such as mechanical strength, porosity, and flexibility, are critical in supporting the wound bed and facilitating cell migration [101]. However, achieving the optimal balance of mechanical properties and biochemical cues to mimic the native ECM remains a challenge, as it requires precision in replicating the complex environment that promotes cellular functions and tissue regeneration. Future research should focus on tailoring mechanical properties, developing biomaterials with tunable mechanical properties to match the dynamic requirements of different phases of wound healing [102]. Chemical factors play a crucial role in modulating the immune response and promoting tissue regeneration [103]. In the wound-healing process, the immune microenvironment evolves, necessitating immune-regulating biomaterials that can dynamically adapt to these shifts, exhibiting distinct antibacterial mechanisms and applications at various stages of infection [104]. The biological response to biomaterials is influenced by the complex interactions between the materials and the host’s immune system [105]. Addressing these challenges requires understanding and optimizing the interactions between biomaterials and different cell types involved in wound healing to enhance integration and tissue regeneration, as well as the design of biomaterials that can modulate the immune response to promote a pro-healing environment while preventing infection [106].

The complexity of wound healing necessitates a multidisciplinary approach, integrating insights from materials science, immunology, and clinical medicine. While significant progress has been made in the development of immunomodulatory biomaterials for wound healing, challenges persist. The future of this field lies in addressing these challenges through innovative research, interdisciplinary collaboration, and a commitment to translating findings into clinical practice. By doing so, we can unlock the full potential of biomaterials to improve wound-healing outcomes and enhance patient care.

## Figures and Tables

**Figure 1 bioengineering-11-01017-f001:**
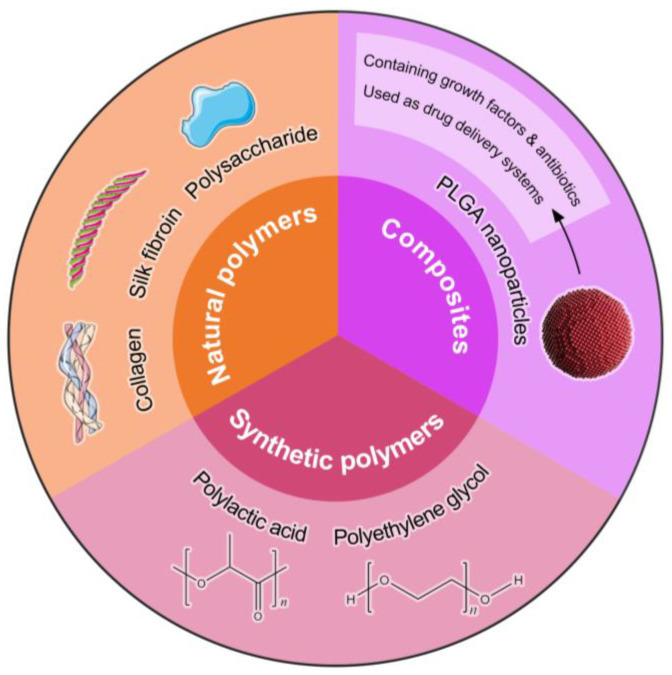
This diagram categorizes immunomodulatory biomaterials into three primary groups: natural polymers, synthetic polymers, and composites. Natural polymers include biocompatible materials such as collagen, gelatin, and silk fibroin, which support cellular activities and tissue regeneration. Synthetic polymers, such as polylactic acid (PLA) and polyethylene glycol (PEG), offer tunable properties for drug delivery and scaffold design. The composite section highlights materials that integrate both natural and synthetic components, exemplified by PLGA nanoparticles, which can encapsulate growth factors and antibiotics, enhancing their efficacy as drug delivery systems in wound-healing applications. This classification underscores the diverse approaches in developing advanced biomaterials for improved therapeutic outcomes.

**Figure 2 bioengineering-11-01017-f002:**
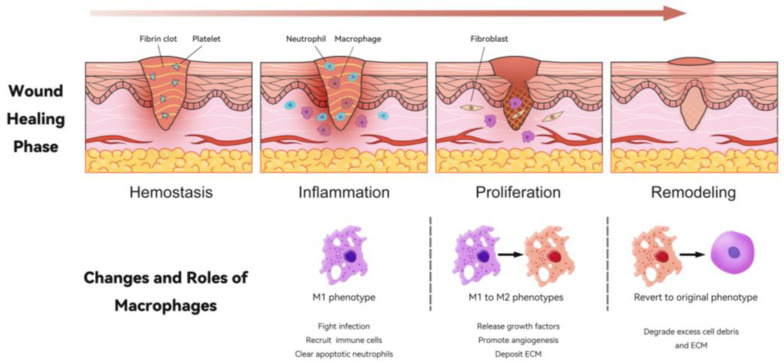
Changes in and roles of macrophages during the wound-healing process. This illustration depicts the four key phases of wound healing: hemostasis, inflammation, proliferation, and remodeling. In the hemostasis phase, a fibrin clot forms, which is essential to stop bleeding. Following this, during the inflammation phase, neutrophils and macrophages are recruited to the wound site; macrophages primarily adopt the M1 phenotype, which is critical for fighting infection and clearing apoptotic neutrophils. In the proliferation phase, macrophages transition from M1 to M2 phenotypes, releasing growth factors, promoting angiogenesis, and depositing extracellular matrix (ECM). Finally, in the remodeling phase, macrophages can revert to their original phenotype (M0), aiding in the degradation of excess cell debris and ECM, thus promoting efficient tissue restoration. This dynamic interplay of macrophages is crucial for successful wound healing.

**Figure 3 bioengineering-11-01017-f003:**
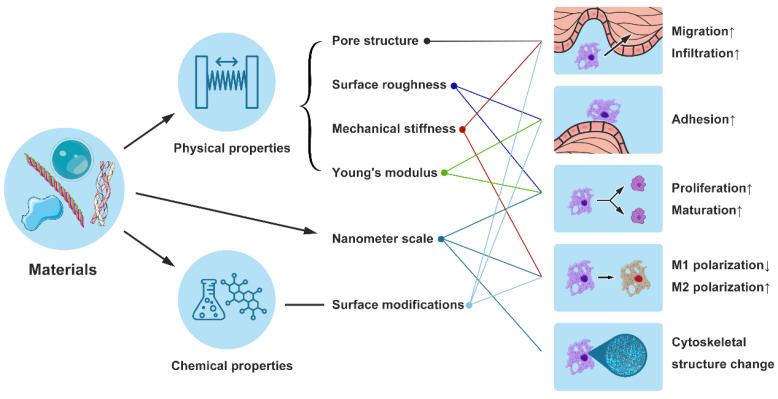
This figure illustrates the design and mechanisms of immunomodulatory biomaterials that affect macrophage behavior. Left Section: Depicts the impact of physical (pore structure, surface roughness, mechanical stiffness, Young’s modulus) and chemical properties (surface modifications) of biomaterials on macrophage functions such, as migration and infiltration. Middle Section: Highlights the role of nanometer-scale materials (e.g., electrospun nanofibers) in modulating macrophage polarization, promoting M2 polarization while reducing M1 polarization. Right Section: Shows the consequent effects on macrophage behavior, including increased adhesion, proliferation, maturation, and changes in cytoskeletal structure. Upward Arrows: Indicate that specific material properties enhance positive cellular responses. Downward Arrows: Indicate that certain properties may suppress M1 polarization, benefiting overall wound healing by favoring an anti-inflammatory response.

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
