# Peer review of "Development of Biomaterials to Modulate the Function of Macrophages in Wound Healing"

_bioengineering, 2024, doi:10.3390/bioengineering11101017_

Round 1

Reviewer 1 Report

Comments and Suggestions for Authors

This is a very good review manuscript summarizing current knowledge about the role of macrophages in wound healing and their modulation by a variety of biomaterials. It very informative, but also very dense. I think for the information to be of use to other investigators the authors need to add several tables covering individual modulators and their actions and activities. The sections right now contain a lot of good information but it is too much of it in each short paragraph. The title needs to indicate that the biomaterials modulate action and function of macrophages in wound healing, e.g., Development of biomaterials modulating function of macrophages in wound healing. The manuscript is also well written.

Reviewer 2 Report

Comments and Suggestions for Authors

General review of immune modulation of wound healing, but does not specifically discuss new information. The topic should be introduced in more depth for each item and the author's thoughts should also be presented.

Reviewer 3 Report

Comments and Suggestions for Authors

This manuscript is a review. The authors state their document "summarizes how... diverse biomaterials promote wound healing by modulating macrophage behavior" (sic); examine the broader implications of these modulations... (and) discuss the limitations associated with the clinical application of immunomodulatory biomaterials and propose potential solutions" (sic). This is ambitious!

Specific comments.

Ln 16. It is evident that the authors are not well informed: while "Wound healing is a complex and precisely regulated process..."; in humans wound healing does not involve "regeneration". The mechanism of wound healing in humans involves 'tissue repair'. 'Tissue repair' and 'regeneration' are not equivalent, and do not result in equivalent healing outcomes. Please review and revise.

Ln 19. Inconsistency. It is not made clear: are the "phases" of wound healing, and "different stages", the same thing, or are these unrelated? I recommend you be consistent with nomenclature.

Ln 26. Incorrect spelling: "biomatrial" should be 'biomaterial'.

Ln 31. The evidence cited in support of "...wounds are classified as acute and chronic..." is derivative and not widely accepted in the field. Definitions of chronic wounds differ with judication!  Please see e.g.: doi: 10.1089/wound.2015.0635; doi: 10.1016/j.annepidem.2018.10.005.

Ln 32. Grammar. Excessively long sentence with multiple subjects: "Acute wounds... surgical procedures, In contrast... chronic wounds... extending beyond three months". For the authors benefit, the clinical definition of "chronic wounds" is not universal. For example, in the USA, cGCP defines wounds that fail "to progress through the phases of healing in an orderly and timely fashion" within 30 days as chronic; in the UK wounds that fail to close within 12 weeks are chronic. One author even chooses to define a 'wound that lacks a 20–40% reduction in size after 2–4 weeks of optimal treatment, or when there is not complete healing after 6 weeks" is chronic! doi: 10.1111/j.1742-481X.2006.00265.x. Consensus is not evident. Please also see: doi: 10.2340/00015555-2786.

Ln 36. This reviewer interprets the statement "...macrophages exert a crucial influence on the wound healing process within the skin" is misleading for uninformed readers. This statement misrepresents the multiple physiological processes required to effect tissue repair in mammals. Once again, the authors limit themselves to skin; yet "wound healing" is a universal process evident in all tissues! Please review and revise.

Ln 39. For the benefit of uninformed readers, please cite sources for "M1 types involved in pathogen phagocytosis and clearance of damaged cells" and for "M2 types contributinge to reparative and regenerative processes"? (sic)

Ln 41. The sentence "The shift from M1 to M2 polarization signifies the transition from inflammatory to proliferation functions" is uninformative. What is "shift"? What is "...inflammatory...functions"? What is "...proliferation functions"? (sic). Please review and revise.

Ln 42. As a practising wound care physician, I strongly disagree with the statement "wound healing strategies usually focus on macrophages". In the real world, clinical wound care is entirely focussed on achieving wound closure! In humans, wound closure is defined as re-epithelialisation, to which macrophages do not contribute! Macrophages contribute to tissue repair and restoring tissue function. These are processes physiologically distinct to wound healing! Please review and revise.

Ln 43. In order to understand current status of "wound healing process" in humans, I strongly recommend the authors read seminal works on this topic; for example: Singer & Clark (1999) doi: 10.1056/NEJM199909023411006; Eming, Martin, Tomic-Canic (2014) doi: 10.1126/science.276.5309.75; Peña & Martin (2024) doi: 10.1038/s41580-024-00715-1. The objective is tissue repair, not tissue substitute xenographs. 

Please also note that cutaneous wound healing as it occurs in humans is distinct and unlike wound healing as it occurs in almost all other mammal species! For this reason, experimental studies in rodents, and other laboratory animal models rarely recapitulate the process of cutaneous wound healing observed in humans!

Ln 46. It is not made clear how "...their ability to modulate immune responses and promote tissue repair mechanisms" have "...made considerable progress in wound healing research and its clinical applications..."?

I interpret this is merely poor grammar. Nonetheless I do not comprehend the authors argument, here. Please review and revise.

Ln 68. The concept to "...affect immune cell function" does not automatically imply "...thus stimulating tissue healing". (sic) This is overinterpretation. 

Ln 82. What is the evidence "Natural polymers... each contributinge a unique role to the wound healing process"? (sic) Are you referring to extracellular matrices (ECM)?

Ln 88. The point is not made. What is the evidence "physical and chemical properties (of synthetic polymers) are often less controllable, and the range of available options is restricted"? What bearing does this have for mammalian wound healing and tissue repair?

Ln 90. For the benefit of the uninformed reader, please elaborate; what are the "beneficial properties of materials"? (sic)

Ln 101. For the benefit of the uninformed reader, please elaborate; what are the "shortcomings of various synthetic materials used for wound healing are also quite pronounced"? (sic)

Ln 107. This reader does not comprehend the relevance of "exogenous drug delivery system(s)" for mammalian wound healing and tissue repair. 

What is "innovative material processing technologythe internal architecture of the material can be refined, thereby enhancing the therapeutic efficacy of wound healing"? (sic) Please review and revise.

Ln 113. I am uninformed; what is the pertinence of "lotus thread bacterial cellulose hydrogel fiber" (sic) for mammalian wound healing and tissue repair?

What are "high-end surgical sutures"? (sic)

Ln 127. Misspelling. "Wound healings" (sic)

Ln 139. It is not clear to this (expert) reader how precisely, molecular species "IL-6, TNFa and IKL-1b...fight infection"? (sic) Is this not a cellular function/property?

Ln 141. While it is true that myocytes release MCP-1, I suggest the authors update their knowledge of immune cell recruitment. For examples see: doi: 10.1126/article.311733; doi: 10.1155/2012/541471.

What is the evidence monocytes release ROS, MCP-1, MMPs? A citation is necessary.

What is the evidence monocytes recognise "DAMPs... (via) PRRs and sustain inflammation through TLRs and inflammasome activation"? A citation is necessary.

Ln 160. The sentence "...epithelial cells change to reshape the wound" is uninformative. 

Ln 162. It is not clear how "...granulation tissue is progressively replaced by collagen-rich scar tissue with low cellularity"? (sic) Please review and revise.

What is the "...steady state of the wound healing process"? (sic)

Ln 165. This reader does not comprehend the sentence: "Depleted macrophages of mice with closed wound display delay? (sic) Please review and revise.

Ln 168. The participation of macrophages in the wound healing cascade is a continuum; however, is illustrated in Figure 2 as discrete. Of equal concern, is the reappearance of M1 macrophages, aka "original phenotype", during remodelling. This is inaccurate. Does the author mean M0 (not illustrated)? Please review and revise. 

Ln 182. Misspelling. "inno-vative" (sic) is not hyphenated.

Ln 183. Misspelling. "compre-hension" (sic) is not hyphenated.

Ln 186. Misspelling. "Materi-al-based" (sic) is not hyphenated.

Ln 197. Misspelling. "sul-fated" (sic) is not hyphenated.

Ln 201. For the benefit of the uninformed reader, what is "M1/M2 balance"? (sic)

Ln 202. Misspelling. "regeneratio". (sic)

Ln 202. Misspelling. "ure-thane" (sic) is not hyphenated.

Ln 210. Misspelling. "therapeu-tic" (sic) is not hyphenated.

Ln 211. Why is Figure 3 labelled as "Figure 1"? (sic)

I am not certain this Figure is helpful; especially given the number of misconceptions and errors it represents.

Ln 223. What is the evidence "Surface roughness significantly influences macrophage adhesion, proliferation, and maturation processes"? (sic) A citation is necessary.

Ln 227. What is the evidence "Young's modulus... serves as a template for cell growth"? (sic) I suggest this is misinterpretation.

Ln 247. Misrepresentation. "Y et al." (sic); should be 'McWhorter et al.'

Ln 251. I recommend the author’s re-evaluate their understanding of 'live cell-material interactions". While it is true that "This interaction is mediated by adhesion proteins on the cell surface, especially integrins", "physical properties of the extracellular matrix (ECM)" are intermediated by molecular species (usually serum glycoproteins) which passivate material surfaces, creating a bridge between the surface and cell surface receptors. That is, cellular interactions with materials are indirect! See: doi 10.1089/ten.2005.11.1. 

Ln 287. It is not clear to this reader; what are "complex interactions with macrophage responses"? (sic) 

Ln 299. Figure 4. This reader fails to appreciate how this figure adds value to this manuscript's text. Rather than clarifying the authors message, I suggest that it adds confusion. I recommend that it be removed.

Ln 309. The statement "Biomaterials often modulate immune responses..." strongly suggests that at other times, 'Biomaterials do not modulate immune responses'! When are these (not as often?) times? Please review and revise.

Lns 309–318. "4.3.1. Macrophage polarization". This reader does not appreciate how this paragraph adds value to this manuscript. Review, revise and/or remove.

Ln 320. This reader is confused: how do "Hydrogel materials... strides"? (sic) 

Ln 327. "Fu et al. reported that... hydrogels..." requires a citation.

Ln 341. For the benefit of the uninformed reader, please introduce (...at least identify!) microRNA's before discussing "miR-29ab1". I do note that miRs are defined later, in Ln 386!

Ln 355. This reader does not understand why the authors choose to highlight "ROS-scavenging hydrogel... (only for) healing of diabetic wounds"? (sic)

Ln 384. "Extracellular vesicles", are a new reagent; however, has not been previously discussed. This is new information and potentially confuse for the uninformed reader. I am not sure why EVs are introduced here, with SiRNA's, miRNAs, protease inhibitors and cytokines, under the subheading "Composites"? The inclusion of bioactives has been introduced previously.

Consider. Should I interpret that composites are limited to synthetic polymers? 

Ln 388. I am surprised to read the first candidate introduced under "4.3.2.1. Natural polymers" is an "engineered novel hydrogel"! (sic) The polymer, "PADM@MgC", is hardly a "natural polymer"! (sic) Review and revise.

Ln 395. "Shen et al. introduced a novel hydrogel composite..." requires a citation.

Ln 408. A second subsection entitled "Synthetic polymers" ("4.3.2.2. Synthetic polymers"), is reported, just 54 pages after "4.3.1.2. Synthetic polymers" (Ln 354)! Is this an editing widow? For what reason could data reported under 4.3.2.2., not be included under 4.3.1.2.? Is this a simple editing widow, or do the authors have another intention?

Ln 409. The sentence commencing: "Electrospun fibers...", ending with "...tissue regeneration" (Ln 415) is excessively long. It is grammatically poor. Please review and revise.

Ln 416 Undefined abbreviation. What are "M2-exos"? (sic)

Ln 418. Why is "hyaluronic acid" discussed under "Synthetic polymers"? Hyaluronic acid is a natural polymer.

Undefined abbreviation. What is "H2S"? (sic)

Ln 431. Grammar. It is poor grammar to start a sentence using an abbreviation; "PU". This reader finds this especially irritating, when authors us an undefined abbreviation! What is "PU"? (sic)

What is the evidence "PU is a versatile material that 431 can be combined with ZnO nanoparticles to create a composite material with antimicrobial properties"? (sic). A citation is required.

Ln 436. It is not made clear; how does ZnO "lead to more effective management of wound infection and inflammation"? (sic) Wound management is a clinical function; how does ZnO effect clinical practices?

Lns 457–462. The "degradation of biomaterials" is not discussed in this work. Why is the "degradation of biomaterials" introduced as a conclusion?

Comments on the Quality of English Language

The language is acceptable, albeit several examples of poor grammar are evident. Minor sub editorial is required.

Round 2

Reviewer 2 Report

Comments and Suggestions for Authors

The authors rewrote and corrected according to the reviewer's comments. I think this manuscript might be acceptable.

Author Response

Dear Reviewer,

Thank you for your careful consideration of our manuscript. We have made the necessary revisions to the manuscript. We believe that these changes have significantly improved the quality and clarity of our work.

We are grateful for the opportunity to revise our work and believe that our manuscript now provides a valuable contribution to the field of polymeric biomaterials for wound healing applications. We are confident that the manuscript is now acceptable for publication and look forward to your final decision.

Thank you once again for your time and for the constructive feedback provided.

incerely,

Dr. Li

Reviewer 3 Report

Comments and Suggestions for Authors

The authors have accepted most of my suggestions, resulting in a much-improved manuscript. The revised version is more succinct, improved scientifically, and easier to read. Despite the author's diligence in editing this manuscript, several minor issues remain; these I would prefer to be addressed before I recommend it be accepted for publication in Bioengineering. 

Ln 50. Grammar (tense). "advancements" (sic) should be 'advances'

Ln 72. Grammar (syntax). I suggest that "...defined as can regulate..." (sic) be revised. Please replace with (for example): '...defined as (materials that) can regulate...'; or "...defined as (those able to) can regulate..."; or "...defined as (materials able to affect immune responses) can regulate..."; or ""...defined as (materials that can modify immune responses) can regulate...".

Ln 93. The sentences "Synthetic polymers offer high controllability, effectively compensating for the limitations inherent in natural materials. However, their physical and chemical properties are often less controllable, and the range of available options is restricted" are self-contradictory. On the one hand, you claim: "Synthetic polymers (are) control(able)" and "effectively" substitute for the " inherent limitations (of) natural materials". Then on the other hand, you claim: ""Synthetic polymers (are) often less controllable" (sic), and offer "restricted options"! (sic). This does not make sense!

Ln 112. Your argument that "synthetic materials (possess) poor biocompatibility or inappropriate degradation" (sic) is not a sound rationale for creating blended and composite materials. Unless explained, would the non-informed reader appreciate; how does blending mitigate "poor biocompatibility or inappropriate degradation"?

Ln 115. This reader is uninformed; how does "enhancement of material performance or optimization of the composition ratio... facilitate the identification of the most efficacious carrier for an exogenous drug delivery system"? (sic) I do not believe that this question is addressed in subsequent text.

Ln 130. Figure 1. What is the reason "Polysaccharide-based biopolymers, extensively utilized in tissue engineering, drug delivery, and bioelectromechanical systems" (Ln 83) are not included in Figure 1?

Ln 150. Rather than "...neutrophils, and macrophages sequentially arrive at the wound", which implies a passive, potentially accidental mechanism, please revise: "neutrophils, and macrophages" are attracted to the wound site by reactive oxygen species (ROS) released with injured tissue. Importantly, this is an active mechanism! The release of proinflammatory cytokines by inflammatory cells is subsequent to this early physiological event. Please see: Enyedi & Niethammer. 2015. doi: 10.1016/j.tcb.2015.02.007. 

Ln 157. I think that you will find "neutrophils (and) macrophages" 'clear' apoptotic cell debris via phagocytosis. Pinocytosis, a different mechanism, is used primarily to 'clear' (engulf) fluid-phase macromolecules. For accuracy, I could add further complication, mentioning 'necroptosis', ferroptosis', and 'pyroptosis'; further mechanisms used by "neutrophils (and) macrophages" to engulf extracellular species! Let us keep this simple and report phagocytosis, please?

Ln 159. Grammar. With; "granulation tissue forms alongside new blood vessels through angiogenesis" the authors did not intend to state that granulation tissue "forms... through angiogenesis" (sic), a property of endothelia. Please review and revise, to clarify that granulation tissue development de novo, is not by angiogenesis. Furthermore, the growth of blood vessels via angiogenesis is subsequent to granulation; it does not "form alongside... blood vessels" (sic)! Please review and revise.

Ln 165. Grammar. I recommend "macrophages shift from M1 to M2 phenotypes" be revised. As a verb, "shift" means "cause to move from one place to another". In the current context, I suggest this is not correct. I suggest the M1 to the M2 phenotype is a polarization, or transition, event. Please review and revise. Please see doi: 10.1038/s41392-023-01452-1.

The current description of macrophage phenotypes is binomial. However, macrophage phenotypes are a continuum. This is challenging to illustrate; and reason many others use terms 'M1-like', 'M2-like', etc.

Ln 172. For clarity, I recommend specifying that when "macrophages revert to their original phenotypes" this is the M0/M1-like phenotype. In Figure 2 (Ln 181) this "original phenotype" is illustrated with a unique, un-named cell phenotype! (i.e. not the original M1 phenotype!)

Ln 213. In my opinion, "...stiffness, can also modulate the polarization of macrophages" is overstatement. I prefer 'modify', 'modulate', 'regulate'. In other words, macrophages responses are not determined solely by the "mechanical properties of scaffolds". Like all mammalian cells, macrophages sense and respond to their whole immediate microenvironment.

Ln 216. Please include source/citation for "Guo et al."?

Ln 233. Contrary to the legend, Figure 3 does not illustrate "...effective wound healing". (sic)

Ln 241. Would the uninformed reader know; what is "Young's modulus"? (sic)

Note. "Young's modulus" is a parameter not limited to "the field of tissue engineering"! (sic) Please review and revise.

Ln 257. Please include source/citation for "Almeida et al."?

Ln 306. What is the evidence "During the initial phase of wound healing, the predominant macrophage phenotype is M1"? (sic) A citation is required.

Ln 336. What is the evidence "The GM-P@HA-P hydrogel... enhances chronic wound healing"? (sic) A citation is required.

Ln 341. Please include source/citation for "Zhang et al."?

Ln 346. Would the uninformed reader know; what is "diabetic inflammation"? (sic)

Ln 360. Grammar. As currently written, "(ROS)-scavenging hydrogel... accelerat(e)ing the healing of diabetic wounds". I suspect the authors did not intend to limit this material to diabetic wounds.

Ln 406. Undefined abbreviation: "DPLG". (sic)

Ln 412. What is the evidence "Electrospun fibers facilitate wound healing by releasing IL-10 in a controlled manner"? (sic) What is the evidence demonstrating "Electrospun fibers" trigger this universal response?

Would the uninformed reader know; "they release a low dose..."? (sic) Who are "they"? 

We have read about M2 macrophages; however, what is "an M2c macrophage response"? (sic)

Would the uninformed reader know; who are "they increase IL-10 release and work with M2c cytokines"? (sic) What is "work"? (sic)

Ln 416. What is "organized tissue regeneration"? (sic)

Ln 457. Incomplete sentence: "...achieving the optimal balance to mimic the native extracellular matrix (ECM) remains a challenge." (sic) Would the uninformed reader know; "optimal balance... of what?

Ln 461. The "degradation of biomaterials play a crucial role" (sic) is not discussed in this manuscript! It is not appropriate to introduce new subjects in the conclusion!

Comments on the Quality of English Language

While it is evident English is not the authors first language, the text is readable. However, I do recommend review and revision be applied to multiple instances of poor grammar.

Author Response

请参阅附件
